# Norepinephrine May Oppose Other Neuromodulators to Impact Alzheimer’s Disease

**DOI:** 10.3390/ijms22147364

**Published:** 2021-07-08

**Authors:** Paul J. Fitzgerald

**Affiliations:** Department of Psychiatry, University of Michigan, Ann Arbor, MI 48109, USA; pfitz1940@gmail.com

**Keywords:** neurodegeneration, dementia, cognitive impairment, noradrenaline, locus coeruleus, clonidine, guanfacine, propranolol, prazosin, terazosin, cholinesterase inhibitors

## Abstract

While much of biomedical research since the middle of the twentieth century has focused on molecular pathways inside the cell, there is increasing evidence that extracellular signaling pathways are also critically important in health and disease. The neuromodulators norepinephrine (NE), serotonin (5-hydroxytryptamine, 5HT), dopamine (DA), acetylcholine (ACH), and melatonin (MT) are extracellular signaling molecules that are distributed throughout the brain and modulate many disease processes. The effects of these five neuromodulators on Alzheimer’s disease (AD) are briefly examined in this paper, and it is hypothesized that each of the five molecules has a u-shaped (or Janus-faced) dose-response curve, wherein too little or too much signaling is pathological in AD and possibly other diseases. In particular it is suggested that NE is largely functionally opposed to 5HT, ACH, MT, and possibly DA in AD. In this scenario, physiological “balance” between the noradrenergic tone and that of the other three or four modulators is most healthy. If NE is largely functionally opposed to other prominent neuromodulators in AD, this may suggest novel combinations of pharmacological agents to counteract this disease. It is also suggested that the *majority* of cases of AD and possibly other diseases involve an excess of noradrenergic tone and a collective deficit of the other four modulators.

## 1. Introduction

The discovery of the detailed double helix structure of DNA by Franklin, Watson, Crick and colleagues in the early 1950s played a critical role in the birth of modern molecular biology [1,2]. Soon afterwards, the “central dogma” of molecular biology was put forth, which describes how DNA is transcribed into (messenger) RNA, and RNA is translated into protein [3]. This model has formed a cornerstone of our understanding of intracellular physiology, and has led to much progress in describing how a variety of diseases affect cells. Much of modern biomedical research, largely conducted since that time, has indeed focused on intracellular molecular processes. Although it should be acknowledged that gaining a greater understanding of molecular processes inside of cells is critical for understanding a wide range of disease processes, in this paper it is suggested that understanding the perhaps more limited set of extracellular signaling molecules that modulate intracellular processes, is also critical for delineating many disease processes.

The current paper focuses on the potential role of five key neuromodulators in Alzheimer’s disease (AD): norepinephrine (NE), serotonin (5-hydroxytryptamine, 5HT), dopamine (DA), acetylcholine (ACH), and melatonin (MT). These five small molecules are distributed throughout the brain and also widely in the periphery, and they appear to modulate many psychiatric, neurologic, and peripheral disease processes [4,5,6,7,8,9,10]. In this paper, it is suggested that NE may be largely functionally opposed to 5HT, ACH, MT, and possibly DA in the pathophysiology underlying AD. Although several theoretical publications and a growing body of data support the possibility that *elevated* NE may cause or worsen the majority of cases of AD, other data suggest that *decreased* noradrenergic signaling may be a causative factor in some cases (e.g., [4,5,11,12]).

In this paper, it is also suggested that partially through its interaction with the neurotransmitters glutamate and GABA, NE may be biased toward being an “excitatory” neuromodulator, in a variety of circuits and synapses. NE may partially achieve this effect through facilitation of glutamatergic signaling and relative suppression of GABAergic transmission, while also producing “excitation” through direct activation of noradrenergic receptors (or a subset of them). In contrast, 5HT, ACH, MT, and in some instances DA may be biased toward “inhibition” and principally antagonize the physiological effects of NE. These three modulators may tend to enhance GABAergic signaling while dampening that of glutamate, while also being “inhibitory” through activation of (a subset of) their own receptors. Furthermore, in this paper it is hypothesized that each of the five neuromodulators has a u-shaped (or Janus-faced, see below) dose-response curve, wherein too little or too much signaling is pathological in AD and possibly other diseases. In this scenario, physiological “balance” between NE and the other three or four neuromodulators is most healthy. While it is not clear from the biomedical literature that NE, relative to the other four modulators, has opposing effects on prominent *intracellular* signaling pathways such as Ras/MAPK, JAK/STAT, and PI3K/Akt, one possibility is that functional opposition is achieved through interaction with different cell types that carry out divergent physiological roles in various organs of the body. It is also suggested here that the *majority* of cases of AD and possibly other diseases may involve an excess of noradrenergic tone, and a collective deficit of the other four neuromodulators. In this scenario, a *minority* of cases of AD may be characterized by a noradrenergic deficit and a relative excess of the other four modulators.

## 2. Relationship with GABA and Glutamate

There is evidence that NE may be biased toward “excitation”, at least in a subset of its circuits, by enhancing glutamatergic signaling and/or suppressing GABAergic transmission [13,14,15]. These data on NE are based in both *in vivo* and *in vitro* animal studies of the locus coeruleus, as well as pharmacological studies with the noradrenergic alpha2 agonist drug, dexmedetomidine [13,14,15]. In contrast, 5HT may be biased toward “inhibition” by enhancing GABAergic transmission and/or suppressing glutamatergic signaling, based on *in vitro* data including the SSRI citalopram [16,17]. This is not to suggest that there are no exceptions, in various circuits and synapses, to this heuristic model of NE and 5HT.

DA may also have inhibitory properties in some instances. Infusion of the DA reuptake inhibitor nomifensine into awake rat medial prefrontal cortex has been associated with increases in synaptic GABA concentration, in the presence of a glutamate reuptake inhibitor [18]. An *in vitro* study of rat entorhinal cortex found that DA reduced glutamatergic signaling more readily than GABAergic signaling, suggesting this neurotransmitter has a tendency to reduce excitation [19]. Infusion of the DA D1 receptor agonist, SKF38393, into medial prefrontal cortex, however, dose-dependently reduced both glutamate and GABA in this brain region [20]. An awake rat microdialysis study suggests that pharmacological activation of D1 or D2 receptors can inhibit local release of glutamate in the medial prefrontal cortex or ventral tegmental area [21].

ACH may also have “inhibitory” signaling characteristics. Systemic administration of the cholinesterase inhibitor donepezil to rats has been shown to increase the extracellular concentration of GABA in the dorsal horn of the spinal cord [22]. In addition, the novel cholinesterase inhibitor ENA713 has been shown to decrease extracellular glutamate in rat hippocampus [23]. The α7 nicotinic acetylcholine receptor antagonist, kynurenic acid, concentration-dependently reduced GABA levels in rat striatum, an effect that was prevented by the cholinesterase inhibitor galantamine [24]. However, systemically administered physostigmine, also a cholinesterase inhibitor, in anesthetized rat increased the concentration of glutamate in the striatum [25]. A possible way to reconcile the discrepancies in these results regarding cholinesterase inhibitors is that their effect on GABAergic or glutamatergic signaling is characterized by a u-shaped or Janus-faced dose-response relationship, where low doses do the opposite of high ones [26].

The neuromodulator MT may also possess “inhibitory” properties. It has been suggested that pineal MT may achieve some of its effects on the central nervous system by modulating GABAergic signaling [27]. Exogenous administration of MT to mice has been shown to increase the concentration of hypothalamic GABA, while not affecting glutamate [28]. 

It should be noted that other factors, in addition to interaction with glutamate and GABA, affect whether a given neuromodulator activates or deactivates a given circuit. For example, while DA through negative feedback regulation can decrease glutamatergic signaling in some cases [29], DA can also act as a pharmacological stimulant throughout the body. Drugs such as amphetamine, cocaine, and apomorphine, for example, which all increase synaptic release of DA, are potent stimulants [30]. This also raises the point that different doses of a drug acting on these neuromodulatory systems (and by extension, the synaptic concentration of the modulators themselves) can produce widely varying effects on neural circuitry and behavior, which relates to u-shaped or Janus-faced dose-response properties that are discussed below [31,32,33]. The suggestion here, that NE and DA may be functionally opposed under some conditions, must be weighed against evidence supporting overlapping effects on brain state, learning, reward processing, and attention [34,35]. NE and DA also synergistically modulate prefrontal functioning through alpha2A and D1 receptors, respectively [36,37]. Perhaps both scenarios, functional opposition versus synergy, exist in disparate circuits and associated divergent behavioral outputs.

An additional caveat to consider, when suggesting that a neuromodulator may be biased toward excitation or inhibition, is that presynaptic and postsynaptic effects of a neurotransmitter may vary depending on concentration, receptor subtype, binding affinity, localization, and the electrical activity of recipient neurons. In contrast to what is suggested above, NE can play an essential role in *decreasing* excitatory transmission [38]. Conversely, the effect of DA on neuronal activity can be either excitatory or inhibitory depending on receptor subtype (e.g., D1 or D2; see below for discussion) and DA concentration [39], where a similar scenario exists for ACH [40]. 

Another way in which these five neuromodulators may interact is through direct agonism of one another’s receptors. For example, NE may activate dopaminergic receptors and DA can activate noradrenergic alpha2 receptors, where the latter phenomenon may serve to inhibit presynaptic NE release and could be a form of functional opposition [41,42,43,44].

The following section briefly discusses how the five neuromodulators described in this manuscript may affect the course of AD.

## 3. Alzheimer’s Disease

Cholinesterase inhibitors, which boost synaptic ACH, are the major class of drugs currently used to treat AD, although their clinical efficacy has been questioned [45]. Similarly, degeneration of the cholinergic basal forebrain, a structure which transmits ACH throughout the brain, is strongly implicated in the progression of AD [46,47]. Thus, these pharmacological and neuroanatomical data implicate ACH in this neurologic disorder.

The locus coeruleus, which provides NE input to widespread regions of the brain, has repeatedly been shown to degenerate in AD [48], suggesting a noradrenergic deficit in this disorder. A previous publication has suggested, however, that *elevated* noradrenergic signaling may in some cases be an etiological factor in AD [11], and more recent papers have also suggested that synaptic NE may be elevated or at least not diminished in spite of degeneration of the locus coeruleus, especially in early stages of the disease [4,5,12]. Regarding molecular pathophysiological mechanisms of NE in this disease, it has been shown that amyloid precursor protein (APP) may counteract desensitization and internalization of the noradrenergic alpha2A receptor [49]. This could suggest that the formation of amyloid plaques is associated with counteracting elevated noradrenergic signaling in AD (or its prodromal state) by activating alpha2A autoreceptors, which then function to decrease presynaptic release of NE. A recent study of genetically modified mice, which had the APP family of molecules knocked out, revealed hyperexcitability of hippocampal neurons via Kv7 channels [50]. One possibility is that NE is also elevated in these animals, contributing to hyperexcitability through a yet to be described molecular mechanism, consistent with a theme in this paper suggesting NE is biased toward “excitation”. Beta amyloid is also capable of inducing internalization and degradation of beta2 adrenergic receptors [51], suggesting another way that amyloid plaques may counteract elevated noradrenergic signaling. 

Another molecular insight from one of these research groups is the recent finding that beta amyloid, by binding to an allosteric site on the alpha2A receptor, redirects noradrenergic signaling to promote tau hyperphosphorylation, which may contribute to degeneration of locus coeruleus neurons [52]. A third insight from an earlier study by this group may suggest that activation of the alpha2A receptor by NE can increase production of beta amyloid [53]. These two studies and those reviewed in the previous paragraph are collectively consistent with elevated noradrenergic signaling playing an etiological role in AD, and may also partially explain why compounds designed to remove amyloid plaques do not have therapeutic properties in this disease. It is suggested here that the formation of amyloid plaques in association with AD (or even in the absence of either AD or cognitive impairment) may be a compensatory mechanism in the brain for counteracting long-term elevated noradrenergic signaling and associated neural hyperexcitability. Degeneration of the locus coeruleus may itself be a late-acting compensatory mechanism for counteracting this long-term elevated noradrenergic tone. The observation that pharmacological activation of beta2 adrenergic receptors (as well as NE activating alpha2A receptors [53]) facilitates the production of beta amyloid [54], which may suggest that endogenous NE itself leads to increased beta amyloid, is consistent with the existence of a negative feedback loop in which elevated NE generates more of these plaques which in turn tends to suppress the elevated noradrenergic signaling.

The neuromodulator 5HT may also play a role in Alzheimer pathophysiology and symptomatology. AD treatment with the SSRI citalopram can result in improvements in confusion, irritability, anxiety, depression, and restlessness [55]. The SSRI sertraline is also effective, and superior to a placebo, at reducing depressive symptomatology in this disease [56]. An additional study of sertraline in major depression accompanying AD, found that this drug reduced depression, lessened behavioral disturbance, and improved activities of daily living [57]. Finally, sertraline was superior to the noradrenergic tricyclic desipramine in treating cognitive, depressive, and behavioral symptoms [58]. A study of comorbid major depression and AD found that the SSRI fluoxetine and the (at least partially noradrenergic boosting) drug amitriptyline were similarly effective in treating depressive symptoms, but fluoxetine was much better tolerated [59]. In individuals with mild cognitive impairment, which may be a prodromal state for AD, fluoxetine improved Mini-Mental State scores, as well as immediate and delayed logical memory scores [60]. Clomipramine, a 5HT boosting tricyclic antidepressant, has been shown to improve mood in probable AD, although it has also been associated with a decrease in Mini-Mental State score [61]. Whether diminished serotonergic signaling really is an underlying etiological factor in AD (as opposed to mediating comorbid depression-related symptoms only) remains to be determined. These data on 5HT nonetheless suggest that boosting this neuromodulator may have therapeutic effects on the symptomatology of AD.

Dopaminergic pharmacological agents also appear to modulate symptoms associated with AD. Apathy, which may be present in up to 70 percent of individuals with AD, is responsive to the dopaminergic stimulant, methylphenidate [62,63]. Apathy in AD is also associated with a muted subjective response to the psychostimulant dextroamphetamine, suggesting dysfunction in dopaminergic reward circuitry [64]. Double-blind, placebo-controlled administration of methylphenidate in AD is associated with improved functioning, cognition, caregiver burden, and depression [65]. Similarly to 5HT, it remains to be determined whether these dopaminergic agents are only affecting comorbid depressive symptoms in AD.

MT supplementation may also be therapeutic in AD. Daily administration for 24 weeks of prolonged-release MT, in a double-blind placebo-controlled fashion, to individuals with AD helped with sleep maintenance and cognitive functioning [66]. Another (double-blind) study also found that MT improved sleep time in individuals with AD [67]. Moreover, daily MT supplementation for four months improved sleep disturbances and sundowning, where the latter term describes worsening of symptoms during the late afternoon or evening [68]. MT modulates beta amyloid signaling in human neuroblastoma cells in a manner that may counteract pathophysiological processes leading to AD [69], and protective effects of MT against beta amyloid excitotoxicity in chick retinal cells may be mediated by activation of GABAergic receptors [70].

The data summarized above on AD are in many cases consistent with the hypothesis that elevated noradrenergic signaling promotes AD, whereas pharmacological boosting of ACH, 5HT, MT, and possibly DA may counteract its deleterious symptoms.

## 4. U-Shaped or Janus-Faced Dose-Response Curves

This paper proposes that NE may be largely functionally opposed to 5HT, ACH, MT, and in some cases DA, in AD and possibly other diseases. Another important aspect of these five modulators to consider is that each may exhibit a u-shaped or Janus-faced dose-response curve in a variety of physiological processes. On the one hand, a u-shaped relationship implies that there is an “optimal” signaling level of each modulator for a given process, wherein higher or lower amounts of signaling do not produce the therapeutic effect. A Janus-faced relationship, on the other hand, suggests that, although a certain low or moderate amount of transmission is optimal, a high level of transmission is actually aversive or toxic. It is suggested here that determining the conditions under which these modulators exhibit u-shaped or Janus-faced properties, including at a molecular or receptor level, is an open question for future experimental investigation. There are already some studies suggesting u-shaped or Janus-faced relationships for all five neuromodulators described here [26,31,32,71,72,73,74,75,76].

## 5. Evaluation of the Hypothesis

The studies reviewed above on AD suggest that all five modulators can affect the manifestation of the disease, and genetic or other variation in these systems may indeed contribute to the etiology of AD. It is less established whether each neuromodulator *bidirectionally* affects AD, wherein relatively low versus relatively high amounts of signaling have opposite effects on AD (i.e., therapeutic versus exacerbating effects). If there is bidirectional modulation, this would be consistent with u-shaped or Janus-faced dose-response curves for each modulator.

To directly address the overarching hypothesis put forth in this paper, additional animal and human subjects studies should be carried out. In animals, perhaps especially in rodents, pharmacological agents should be used to determine: (1) how increasing or decreasing signaling within a given modulatory system affects glutamatergic versus GABAergic transmission (in combination with microdialysis); (2) whether noradrenergic signaling tends to be functionally opposed to the other three or four modulators; (3) whether each modulator has u-shaped or Janus-faced dose-response properties; (4) how the Ras/MAPK, JAK/STAT, PI3K/Akt pathways are altered in a given circuit or cell type; and (5) how these modulators affect disease genesis or progression in animal models of AD. These five general questions can also be addressed through chemogenetic and optogenetic approaches, which are informative for investigating causation and cell-type specificity. Additional human subjects studies should include: (1) more prospective clinical trials of pharmacological agents modulating AD; and (2) additional pharmacological and genetic retrospective epidemiological analysis of medical records in the context of AD.

## 6. Consequences of the Hypothesis

Perhaps the most important consequence, if the main hypothesis (i.e., functional opposition) is true, is that it would suggest novel pairs (or perhaps triplets) of existing drugs that could be used to more effectively prevent or treat AD. For example, on the one hand, if *most* cases of AD involve a relative excess of NE and a relative deficiency of 5HT, DA, ACH, and MT, then a noradrenergic transmission reducing drug (clonidine, guanfacine, dexmedetomidine, propranolol, carvedilol, nebivolol, prazosin, doxazosin, or terazosin) could be paired with a 5HT boosting drug (SSRIs, phenelzine, tranylcypromine, or possibly clomipramine), or an ACH boosting compound (cholinesterase inhibitors), or MT. If, on the other hand, a *minority* of cases of AD are characterized by a deficiency of NE and an excess of the other four modulators, this would suggest a different pharmacological treatment strategy: boosting noradrenergic transmission (desipramine, nortriptyline, reboxetine, or atomoxetine) while suppressing transmission of the other modulators (e.g., with ACH blocking agents such as atropine, scopolamine, or mecamylamine).

If NE really is in many cases functionally opposed to other neuromodulators, this would have additional consequences. It would stand in contrast to the hypothesis that each of these modulators has a rather distinct set of functions (e.g., for NE: stress response, alertness, and memory functionality; for 5HT: cognitive flexibility and digestion; for DA: learning, reward, and movement) [26]. Perhaps both scenarios are partially correct. Previous studies have already stated that NE and 5HT may be largely functionally opposed [77,78]. NE and ACH are known to be more explicitly functionally opposed in the periphery, at the output of the sympathetic and parasympathetic branches, respectively, of the autonomic nervous system. Janowsky et al. (1972), by putting forth the cholinergic-adrenergic hypothesis of mood disorders, suggested that ACH and NE are functionally opposed in the brain, at least in mood-related circuits [79].

Although it is suggested here that NE is biased toward “excitation”, and 5HT, ACH, MT, and in some cases DA may be biased toward “inhibition”, this refers to the overall effect of each transmitter system on neural function. In this scenario, the receptor subtypes within each modulatory system may be either excitatory or inhibitory, which may in part map onto their G protein coupled receptor (GPCR) subtype, where Gs and Gq are largely excitatory, and Gi is inhibitory. Noradrenergic alpha1, beta1, beta2, and beta3 receptors would be excitatory, and alpha2 would be inhibitory. For example, serotonergic 5HT2A would be excitatory, whereas 5HT1A and 5HT2C would be inhibitory. In this scenario, D1-like DA receptors would be excitatory, and D2-like DA receptors would be inhibitory, and so forth, for ACH (with muscarinic receptors typically being inhibitory and ionotropic nicotinic receptors being largely excitatory) and MT (where MT1 and MT2 are inhibitory).

If the five extracellular neuromodulators described in this paper play critical roles in regulating intracellular molecular pathways, what controls these five modulators? While this is a subject for other investigations, genetic, epigenetic, and environmental factors very likely affect both the tonic and phasic signaling properties of these five molecules. Top-down psychological influences, including an individual’s reactivity to psychosocial stress, may also affect their signaling properties in diverse brain circuits and in the periphery. For example, these neuromodulators may be affected by depressive rumination (recursive self-focused thinking), which is associated with recruitment of limbic, medial, and dorsolateral prefrontal regions including the default mode network [80,81].

NE may also largely serve to activate a wide range of physiological processes, many of which are not well characterized at this time, including: embryonic development (and possibly postnatal development, leading up to adulthood), consistent with the NE-lowering HDAC inhibitor valproic acid having teratogenic properties [82]; noradrenergic promotion of hunger, feeding, and weight gain or loss, including a potential relationship with cachexia in cancer; flashbulb memories and learning associated with surprise [83] which historically have been more attributed to DA, where beta adrenergic and DA D1 receptors may have differential effects [84]; tardive dyskinesia, L-DOPA induced dyskinesia, and neuroleptic malignant syndrome [85]; postural sway associated with schizophrenia and other disorders; locomotion stimulating NE release and NE release stimulating locomotion, with a neural correlate of enhanced theta (and possibly gamma) oscillations; and focusing the “spotlight” of attention, including spatial attention focused on different areas of one’s body, where NE plays a role in pupillary dilation during orienting of attention [86]. It is suggested here that this general “activational” role for NE, in the brain and periphery, is not limited to functional opposition of the parasympathetic nervous system by sympathetic signaling.

In conclusion, this paper has suggested that NE occupies a unique position among other neuromodulators, in that it is in many cases functionally opposed to 5HT, ACH, MT, and in some instances DA. Furthermore, each of these five modulators may exhibit u-shaped or Janus-faced dose-response properties, and they also appear to play an important role in AD. It is suggested here that the majority of AD cases involve elevated noradrenergic signaling and a collective deficit of the other three or four modulators (if DA is included). Most importantly, the opposing relationships described here suggest novel combinations of pharmacological agents to prevent or treat AD and possibly other diseases. Future preclinical and clinical studies should now further investigate these translationally relevant topics.

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
