# Peer review of "Norepinephrine May Oppose Other Neuromodulators to Impact Alzheimer’s Disease"

_ijms, 2021, doi:10.3390/ijms22147364_

Round 1

Reviewer 1 Report

Manuscript entitled:”Norepinephrine may oppose other neuromodulators to hierarchically control diverse signaling pathways in health and disease” contains the interesting hypothesis that norepinephrine is an “excitatory” neurotransmitter and serotonin, dopamine, acetylcholine, and melatonin are “inhibitory” neurotransmitters.

The Author discusses the action of as many as 5 neurotransmitters and 4 different diseases. Due to such a large number of factors discussed, the discussion of this hypothesis is very cursory. There are too many variables in this version of the manuscript and therefore it is very difficult to read.

 I think it would be better to focus on 2 neurotransmitters (e.g. norepinephrine and dopamine or norepinephrine and serotonin) and discuss their opposing effects in the brain in more detail in various diseases. Possibly, the effects of all 5 neurotransmitters should be discussed only in 1 disease. I believe that the Author has not proved his hypothesis; in particular, the chapters "Major depression", "Epilepsy", "Alzheimer's disease" and "Cancer" contain only the effects of individual neurotransmitters - there is no evidence to confirm the author's hypothesis.

Furthermore, I disagree with the statement that dopamine is an inhibitory neurotransmitter. It is true that under physiological conditions, dopamine acts as a feedback loop with glutamate, i.e. an increase in glutamate release causes a decrease in dopamine release, and vice versa, i.e. an increase in dopamine release causes a decrease in glutamate release. But the increase in dopamine release causes a strong stimulation of the body! This is best seen after the administration of stimulants such as amphetamines, cocaine, apomorphine, etc .. Therefore, classifying neurotransmitters only on the basis of their effect on glutamate and/or GABA is a great simplification and does not reflect the actual effect on brain function. An important aspect is also the different action of drugs depending on the dose used. e.g. ketamine (NMDA receptor inhibitor) in low doses has an antidepressant effect, in medium doses it causes memory impairment and hallucinations, while in high doses it is used as an anesthetic.

For the above reasons, I suggest re-editing the manuscript in which the Author should focus on a smaller number of variables.

Author Response

Manuscript entitled:”Norepinephrine may oppose other neuromodulators to hierarchically control diverse signaling pathways in health and disease” contains the interesting hypothesis that norepinephrine is an “excitatory” neurotransmitter and serotonin, dopamine, acetylcholine, and melatonin are “inhibitory” neurotransmitters.

The Author discusses the action of as many as 5 neurotransmitters and 4 different diseases. Due to such a large number of factors discussed, the discussion of this hypothesis is very cursory. There are too many variables in this version of the manuscript and therefore it is very difficult to read.

I think it would be better to focus on 2 neurotransmitters (e.g. norepinephrine and dopamine or norepinephrine and serotonin) and discuss their opposing effects in the brain in more detail in various diseases. Possibly, the effects of all 5 neurotransmitters should be discussed only in 1 disease. I believe that the Author has not proved his hypothesis; in particular, the chapters "Major depression", "Epilepsy", "Alzheimer's disease" and "Cancer" contain only the effects of individual neurotransmitters - there is no evidence to confirm the author's hypothesis.

Response: I have now consolidated the manuscript to focus on only one disease (Alzheimer’s; AD), but still describe the potential effects of all 5neurotransmitters in AD. This makes the manuscript much more focused now, thank you. Some evidence for functional opposition between NE and 5HT, ACH, MT (and possibly DA) is presented in the section on AD, in that elevated NE may promote symptomatology, whereas pharmacological boosting of the other 4 modulators may be therapeutic in this disease.

Furthermore, I disagree with the statement that dopamine is an inhibitory neurotransmitter. It is true that under physiological conditions, dopamine acts as a feedback loop with glutamate, i.e. an increase in glutamate release causes a decrease in dopamine release, and vice versa, i.e. an increase in dopamine release causes a decrease in glutamate release. But the increase in dopamine release causes a strong stimulation of the body! This is best seen after the administration of stimulants such as amphetamines, cocaine, apomorphine, etc .. Therefore, classifying neurotransmitters only on the basis of their effect on glutamate and/or GABA is a great simplification and does not reflect the actual effect on brain function. An important aspect is also the different action of drugs depending on the dose used. e.g. ketamine (NMDA receptor inhibitor) in low doses has an antidepressant effect, in medium doses it causes memory 
impairment and hallucinations, while in high doses it is used as an anesthetic.

Response: These are valid criticisms. I have now suggested, throughout the manuscript, that dopamine (DA) may only have inhibitory properties under some conditions. The following general statement has now been added at the end of the section on GABA and glutamate: “It should be noted that other factors besides interaction with glutamate and GABA affect whether a given neuromodulator activates or deactivates a given circuit. For example, while DA through negative feedback regulation can decrease glutamatergic signaling in some cases 28, DA can also act as a pharmacological stimulant throughout the body. Drugs such as amphetamine, cocaine, and apomorphine, for example, which all increase synaptic release of DA, are potent stimulants 29. This also raises the point that different doses of a drug acting on these neuromodulatory systems, (and by extension, the synaptic concentration of the modulators themselves) can produce widely varying effects on neural circuitry and behavior, which relates to u-shaped or Janus-faced dose-response properties that are discussed below 30–32.”

For the above reasons, I suggest re-editing the manuscript in which the Author should focus on a smaller number of variables.

Response: The manuscript is now thoroughly revised and much more focused. Thank you.

Reviewer 2 Report

It is a well-written paper that claims a general oppose interaction between neuromodulators. Specifically of norepinephrine vs. DA, 5HT, ACH, and MT, assigning opposite roles in several central and functions based on well-selected experimental data supporting the hypothesis. The main impact of the antagonistic interaction occurs in pathological conditions by an imbalance between excitatory (NE) vs. inhibitory modulators, and four cases are presented. However, the cellular mechanism remains to be elucidated. Nevertheless, from my perspective, this concern does not significantly limit the validity of the hypothesis. As a commentary, some reports present evidence that neuromodulators, especially dopamine and NE, can share the ability to bind the receptor of the other neuromodulator (Root et al., 2015. J Neurosci, 35, 3460; Aslanoglou et al., 2021 Translational Psychiatry 11:59; Sánchez-Soto et al., 2018 Molecular Neurobiology 55:8438; Sanchez-Soto et al., 2016, Molecular Pharmacology 89:457). Thus, this kind of interaction can be involved in the oposse action of neuromodulators?

Author Response

It is a well-written paper that claims a general oppose interaction between neuromodulators. Specifically of norepinephrine vs. DA, 5HT, ACH, and MT, assigning opposite roles in several central and functions based on well-selected experimental data supporting the hypothesis. The main impact of the antagonistic interaction occurs in pathological conditions by an imbalance 
between excitatory (NE) vs. inhibitory modulators, and four cases are presented. However, the cellular mechanism remains to be elucidated. Nevertheless, from my perspective, this concern does not significantly limit the validity of the hypothesis. As a commentary, some reports present evidence that neuromodulators, especially dopamine and NE, can share the ability to bind the 
receptor of the other neuromodulator (Root et al., 2015. J Neurosci, 35, 3460; Aslanoglou et al., 2021 Translational Psychiatry 11:59; Sánchez-Soto et al., 2018 Molecular Neurobiology 55:8438; Sanchez-Soto et al., 2016, Molecular Pharmacology 89:457). 

Response: Thank you. This is a good point about the neuromodulators activating one another’s receptors. The following statement on this topic has been added before the section on AD: “Another way in which these five neuromodulators may interact is through direct agonism of each other’s 
receptors. For example, NE may activate dopaminergic receptors and DA can activate noradrenergic alpha2 receptors, where the latter phenomenon may serve to inhibit presynaptic NE release and could be a form of functional opposition 33–36.”

Reviewer 3 Report

The article's topic is fascinating, as the goal of developing an integrative hypothesis of the kind proposed by Paul J. Fitzgerald, albeit ambitious, could represent a significant advancement. However, in my opinion, the manuscript has serious flaws preventing its publication as is. 
First, it oversimplifies such a complex matter and is therefore potentially misleading. Oversimplification is probably due to the ambition of putting together disparate phenomena and levels of analysis, health, and pathology, from depression to cancer. Moreover, putting together very different pathologies such as depression, epilepsy, Alzheimer's disease, and cancer is somewhat confusing, at least to my reading.
Second, it lacks an adequate appreciation of the vast literature showing that the pre-and post-synaptic effects of almost every neurotransmitter vary, depending on its concentration, receptor subtype, affinity, localization, and electrical activity of recipient neurons, etc.
Statements such as "largely excitatory" or "largely inhibitory" are challenging to sustain based on empirical evidence, as well as of dubious utility, if not perhaps in an introductory neuroscience manual.
NE, for example, also plays an essential role in decreasing excitatory transmission (see PMID: 22717696). Conversely, it has been known for a long time that DA effects on neuronal activity may be both excitatory or inhibitory depending on receptor subtype (D1, D2) and DA concentration (PMID: 2890403). Similarly for acetylcholine (PMID: 229514).
Moreover, while the author hypothesizes an opponent functional relationship between NE and DA, such transmitters show overlapping effects on learning, brain state, reward processing (see PMID: 32038164), and attention. In addition, NE and DA synergistically modulate PFC functioning through α(2A)adrenoceptors and D1-receptors, respectively (PMID: 26790349; 21295057). Such literature must be adequately considered.
Furthermore, the manuscript lacks adequate integration; some parts seem not to respond to a clear logical-discursive function (e.g., the paragraph "Molecular effects of 5HT"). Or at least I don't' catch it. 
In the first paragraph, the author stresses that extracellular signaling controls intracellular processes. Why does he not go further asking what influences extracellular signaling? The reader would expect, or at least I would have expected, to acknowledge that extracellular signaling is driven by the individual's psychological appraisals resulting from brain-environment interaction. This acknowledgment would be highly relevant for major depression and mental disorders in general. Yet, in the paper, there is no reference to the top-down effects of cognitive activity on the neural level, which in my opinion contradicts the introductive claim on central dogma, or at least does not develop it adequately. For example, rumination, or recursive self-focused thinking, an important maintenance factor of depressive episodes, is associated with enhanced recruitment of limbic and medial and dorsolateral prefrontal regions in depression (PMID: 21098808). So, the emphasis on extracellular signaling seems like an unfulfilled promise, a potential not realized.
Finally, several assertions are supported only by self-citation of medical hypothesis papers, which does not seem adequate to me. For instance, when the author affirms that the five neuromodulators considered "appear to modulate many psychiatric, neurologic, and peripheral disease processes" (line 47) (PMID: 19740085; 23410497; 20626335); and again when he claims that "a number of theoretical publications and a growing body of data support the possibility that elevated NE may cause or worsen a majority of cases of these four diseases" (lines 52-54) (PMID: 25271382).
Regrettably, as it is, I think the paper cannot be amended. Instead, I suggest rethinking it from scratch using better support from the literature, focusing and integrating it better, and avoiding forcing a complex functioning in a simplistic hypothesis, even if elegant.

Author Response

The article's topic is fascinating, as the goal of developing an integrative hypothesis of the kind proposed by Paul J. Fitzgerald, albeit ambitious, could represent a significant advancement. However, in my opinion, the manuscript has serious flaws preventing its publication as is. First, it oversimplifies such a complex matter and is therefore potentially misleading. Oversimplification is probably due to the ambition of putting together disparate phenomena and 
levels of analysis, health, and pathology, from depression to cancer. Moreover, putting together very different pathologies such as depression, epilepsy, Alzheimer's disease, and cancer is somewhat confusing, at least to my reading/

Response: I have now focused the manuscript just on Alzheimer’s disease (AD). This helps its readability and makes it much more streamlined; thank you for pointing this out.

Second, it lacks an adequate appreciation of the vast literature showing that the pre-and post-synaptic effects of almost every neurotransmitter vary, depending on its concentration, receptor subtype, affinity, localization, and electrical activity of recipient neurons, etc. Statements such as "largely excitatory" or "largely inhibitory" are challenging to sustain based on empirical evidence, as well as of dubious utility, if not perhaps in an introductory neuroscience manual.
NE, for example, also plays an essential role in decreasing excitatory transmission (see PMID: 22717696). Conversely, it has been known for a long time that DA effects on neuronal activity may be both excitatory or inhibitory depending on receptor subtype (D1, D2) and DA concentration (PMID: 2890403). Similarly for acetylcholine (PMID: 229514).

Response: These are valid points. A paragraph on this topic has been added before the section on AD: “An additional caveat to consider in suggesting that a neuromodulator may be biased toward excitation or inhibition is that presynaptic and postsynaptic effects of a neurotransmitter may vary depending on concentration, receptor subtype, binding affinity, localization, and the electrical activity of recipient neurons. In contrast to what is suggested above, NE can play an essential role in decreasing excitatory transmission 38. Conversely, the effect of DA on neuronal activity can be either excitatory or inhibitory depending on receptor subtype (e.g., D1 or D2; see below for discussion) and DA concentration 39, where a similar scenario exists for ACH 40.”

Moreover, while the author hypothesizes an opponent functional relationship between NE and DA, such transmitters show overlapping effects on learning, brain state, reward processing (see PMID: 32038164), and attention. In addition, NE and DA synergistically modulate PFC functioning throughα(2A)adrenoceptors and D1-receptors, respectively (PMID: 26790349; 21295057). Such literature must be adequately considered.

Response: The following statement has been added on this topic, before the section on AD: “The suggestion here that NE and DA may be functionally opposed under some conditions, must be weighed against evidence supporting overlapping effects on brain state, learning, reward processing, and attention 34,35. NE and DA also synergistically modulate prefrontal functioning through alpha2A and D1 receptors, respectively 36,37.”

Furthermore, the manuscript lacks adequate integration; some parts seem not to respond to a clear logical-discursive function (e.g., the paragraph "Molecular effects of 5HT"). Or at least I don't' catch it.

Response: That paragraph has now been deleted, thank you. Different aspects of manuscript are also more integrated now that the focus is just on AD.

In the first paragraph, the author stresses that extracellular signaling controls intracellular processes. Why does he not go further asking what influences extracellular signaling? The reader would expect, or at least I would have expected, to acknowledge that extracellular signaling is driven by the individual's psychological appraisals resulting from brain-environment interaction. This acknowledgment would be highly relevant for major depression and mental 
disorders in general. Yet, in the paper, there is no reference to the top-down effects of cognitive activity on the neural level, which in my opinion contradicts the introductive claim on central dogma, or at least does not develop it adequately. For example, rumination, or recursive selffocused thinking, an important maintenance factor of depressive episodes, is associated with enhanced recruitment of limbic and medial and dorsolateral prefrontal regions in depression (PMID: 21098808). So, the emphasis on extracellular signaling seems like an unfulfilled promise, a potential not realized.

Response: This is a very good point. While the section on major depression has now been deleted from the manuscript, the topic of what influences the neuromodulators is still relevant, and the following statement has now been added toward the end of the manuscript: “If the five extracellular neuromodulators described in this publication play a critical role in regulating 
intracellular molecular pathways, what controls these five modulators? While this is a subject for other investigations, genetic, epigenetic, and environmental factors very likely affect both the tonic and phasic signaling properties of these five molecules. Top-down psychological influences, including an individual’s reactivity to psychosocial stress, may also affect their signaling properties in diverse brain circuits and in the periphery. For example, these neuromodulators may be affected by depressive rumination (recursive self-focused thinking), 
which is associated with recruitment of limbic, medial and dorsolateral prefrontal regions including the default mode network 73,74.”

Finally, several assertions are supported only by self-citation of medical hypothesis papers, which does not seem adequate to me. For instance, when the author affirms that the five neuromodulators considered "appear to modulate many psychiatric, neurologic, and peripheral disease processes" (line 47) (PMID: 19740085; 23410497; 20626335);

Response: These theoretical, self-citing references for that sentence have now been deleted and replaced by a number of empirical studies or reviews from other researchers.

and again when he claims that "a number of theoretical publications and a growing body of data support the possibility that elevated NE may cause or worsen a majority of cases of these four diseases" (lines 52-54) (PMID: 25271382).

Response: This sentence has now been modified to focus on AD.

Regrettably, as it is, I think the paper cannot be amended. Instead, I suggest rethinking it from scratch using better support from the literature, focusing and integrating it better, and avoiding forcing a complex functioning in a simplistic hypothesis, even if elegant.

Response: The manuscript has now been thoroughly rewritten. It now focuses just on AD, and has eliminated a lot of extraneous information in the process. More details and further discussion have now been added on areas of weakness that were pointed out by all 3 reviewers. The manuscript is now much improved, in my opinion. Thank you.